# Survival of COVID-19 with Multimorbidity Patients

**DOI:** 10.3390/healthcare9111423

**Published:** 2021-10-22

**Authors:** E. Bustos-Vázquez, E. Padilla-González, D. Reyes-Gómez, M. C. Carmona-Ramos, J. A. Monroy-Vargas, A. E. Benítez-Herrera, G. Meléndez-Mier

**Affiliations:** 1Department of Epidemiology, Secretary of Health of the State of Hidalgo, Pachuca C.P. 42182, Hidalgo, Mexico; subdireccion.epidemiologia@ssh.gob.mx (E.B.-V.); epage2090@hotmail.com (E.P.-G.); jmv.semp10@hotmail.com (J.A.M.-V.); 2Public Health Subsecretary of the State of Hidalgo, Pachuca C.P. 42182, Hidalgo, Mexico; subsecretariasaludpublicahgo@gmail.com; 3Directorate of Public Health Policies and Strategies, Secretary of Health of the State of Hidalgo, Pachuca C.P. 42182, Hidalgo, Mexico; concepcion.carmona@ssh.gob.mx; 4Secretary of Health of the State of Hidalgo, Pachuca C.P. 42086, Hidalgo, Mexico; alejandro.benitez@ssh.gob.mx; 5Health Research Coordination, Secretary of Health of the State of Hidalgo, Pachuca C.P. 42083, Hidalgo, Mexico

**Keywords:** SARS-CoV-2 infection, survival rate, hospitalized patients, Hidalgo Mexico, multimorbidity

## Abstract

Background: The outbreak of SARS-CoV-2 abruptly disseminated in early 2020, overcoming the capacity of health systems to respond the pandemic. It was not until the vaccines were launched worldwide that an increase in survival was observed. The objectives of this study were to analyse the characteristics of survivors and their relationship with comorbidities. We had access to a database containing information on 16,747 hospitalized patients from Mexico, all infected with SARS-CoV-2, as part of a regular follow-up. The descriptive analysis looked for clusters of either success or failure. We categorized the samples into no comorbidities, or one and up to five coexisting with the infection. We performed a logistic regression test to ascertain what factors were more influential in survival. The main variable of interest was survival associated with multimorbidity factors. The database hosted information on hospitalized patients from Mexico between March 2020 through to April 2021. Categories 2 and 3 had the largest number of patients. Survival rates were higher in categories 0 (64.8%), 1 (57.5%) and 2 (51.6%). In total, 1741 (10.5%) patients were allocated to an ICU unit. Mechanical ventilators were used on 1415 patients, corresponding to 8.76%. Survival was recorded in 9575 patients, accounting for 57.2% of the sample population. Patients without comorbidities, younger people and women were more likely to survive.

## 1. Introduction

In late 2019, the Wuhan Municipal Health Commission reported a cluster of pneumonia cases of unknown aetiology, with a common source of exposure at the seafood market of Wuhan City, Hubei Province, China [1]. Subsequent investigations identified a new coronavirus as the causative agent of the respiratory symptoms—the Severe Acute Respiratory Syndrome Coronavirus 2 (SARS-CoV-2) was responsible for Coronavirus Disease 2019 (COVID-19) [2]. The disease evolved and spread rapidly, affecting many countries around the globe. By 30 January 2020, Dr. Tedros Adhanom Ghebreyesus, General Director to the World Health Organization (WHO), declared COVID-19 a Public Health Emergency of International Importance (PHEII), issuing recommendations under the International Health Regulations (IHR) [3]. By 11 March 2020, the same organization recognized the SARS-CoV-2 as a new pandemic [4]. Since then, the number of cases worldwide has escalated to around 225 million, from which over 4.5 million have resulted in death and over 200 million have recovered. The countries most affected have been the United States, accounting for 41,741,693 cumulative cases, 677,017 deaths and 31,820,994 patients recovered, followed by India and Brazil accounting for 33,200,877 and 20,974,850 cases and 442,350 and 585,923 deaths, respectively [5]. Mexico, a middle-income country, has accumulated 3,479,999 cases with 266,150 deaths [5], ranking in 15th place of cases and deaths worldwide [5]. As in many other countries, Mexico was struck by the spreading wave, rapidly occupying hospital beds with the sickest patients. COVID-19 demonstrated that the world was less prepared than most had imagined for dealing with a pandemic [6]. Experiences are numerous, and Horwitz et al. [7] evaluated 5121 adults hospitalized with SARS-CoV-2 infection from March through to August 2020 and found that patients hospitalized later during the period were much younger and had fewer comorbidities. Importantly, the authors observed a marked decline in adjusted in-hospital mortality or hospice rates, from 25.6% in March to 7.6% in August. This suggests that more elderly people were hospitalized in the early stages of the pandemic, causing large-scale hospital occupation, with the most vulnerable population bearing comorbidities, who were also more susceptible to death. However, as time went by, there was a shift in the trend of hospitalization towards younger people with less or no comorbidities and more opportunity to survive. Multimorbidity has been recognized by other authors [8,9,10] as an independent risk factor for SARS-CoV-2 outcomes after infection. In a study by Gupta et al. [11], they showed that factors independently associated with death included older age, male sex, higher body-mass index, coronary artery disease, active cancer, and the presence of hypoxemia, liver dysfunction (liver Sequential Organ Failure Assessment score of 2–4 vs. 0: OR, 2.61; 95% CI, 1.30–5.25), and kidney dysfunction (renal Sequential Organ Failure Assessment score of 4 vs. 0: OR, 2.43; 95% CI, 1.46–4.05) at ICU admission. Chudasama et al. [12] compared severe vs. non severe cases of SARS-CoV-2 infection in hospitalized patients and found that clusters of several multimorbidities were more frequent in those with severe SARS-CoV-2 infection. The most common clusters were stroke with hypertension (79% of those with stroke had hypertension); diabetes and hypertension (72%); and chronic kidney disease and hypertension (68%). Multimorbidity was independently associated with a greater risk of severe SARS-CoV-2 infection. In the study reported by Ferroni [10] et al., they identified that survival was higher in younger patients and in females. The negative impact of comorbidities on survival was more pronounced in younger age groups; in the study presented by Iaccarino et al. [13], older age, hypertension, diabetes mellitus, chronic obstructive pulmonary disease, chronic kidney disease, coronary artery diseases, and heart failure were more represented in no survivors than in survivors. ACE (angiotensin-converting enzyme) inhibitors, diuretics, and β-blockers were more frequently used in no survivors than in survivors. Nijman et al. [14] in the Netherlands found no increased mortality risk in male patients, or patients with high BMI or diabetes. In the systematic review developed by Flook et al. [15], age was a prominent predicting factor for death in hospitalized patients with SARS-CoV-2. Age, gender, and comorbidities were commonly assessed as risk factors. The weight of evidence showed increasing age to be associated with severe disease and mortality, and general comorbidities with mortality. Another systematic review by Huang et al. [16] managed to demonstrate in their meta-analysis that diabetes mellitus was associated with both composite poor outcome (RR 2.38 [1.88, 3.03], *p* < 0.001; I(2): 62%) and the subgroup which comprised mortality (RR 2.12 [1.44, 3.11], *p* < 0.001; I(2): 72%), severe COVID-19 (RR 2.45 [1.79, 3.35], *p* < 0.001; I(2): 45%), ARDS (RR 4.64 [1.86, 11.58], *p* = 0.001; I(2): 9%), and disease progression (RR 3.31 [1.08, 10.14], *p* = 0.04; I(2): 0%). Gonzalez Ramirez [17] performed a narrative review of causes of death among COVID-19 patients in Mexico City and found that age, gender, and previous health conditions have a considerable effect on the mortality rate of those confirmed COVID-19 patients, so that older people, men, and people with certain pre-existing health conditions, such as hypertension and diabetes, have a higher risk of death than younger people, women, and people without pre-existing health conditions.

The present study aims to estimate survival and risk factors associated with COVID-19 in multimorbid hospitalized patients.

## 2. Material and Methods

We carried out an observational, retrospective, and analytical study among hospitalized patients from COVID-19 in Hidalgo, Mexico. A confirmatory positive test result to SAR-CoV-2 was obtained by RT-PCR, and the diagnosis, when necessary, was supported with a rapid antigen test or by clinical association. Excluded cases were those lacking incomplete relevant information for the analysis. After cleaning, the total sample consisted of 16,747 hospitalized patients for the study period; however, three hundred and sixty-seven were discarded because of incomplete information.

## 3. Data Collection

We used data from the Epidemiological Case Study of Respiratory Diseases repository database recorded in the Epidemiological Surveillance System of Respiratory Disease platform (ESSRD) from Hidalgo, Mexico. Further, cases reported from 1 March 2020 to 30 April 2021 [1], and the Nominal List of Hospitalized Cases for COVID-19 from the Department of Epidemiology of Hidalgo, Mexico were crosschecked.

For each case, age, sex, occupation, presence of concomitant morbidity (diabetes, hypertension, obesity, CKD, COPD, asthma, immunosuppression, HIV, and heart disease in combination at least with COVID-19), date of onset of signs and symptoms, date of hospital admission, evolution, death if applicable, hospitalized in an Intensive Care Unit (ICU) and whether use of mechanical ventilation was required, were entered into the analysis.

The population admitted to the hospital were patients with different degrees of severity of COVID-19 requiring hospitalization. They signed an informed consent form at the time of admission. The main variable of interest was survival associated with multimorbidity, and as a secondary outcome we assessed other factors identified influencing survival. 

Concomitant morbidities were categorized as follows:

Categories

No additional morbiditiesOne additional morbidityTwo additional morbiditiesThree additional morbiditiesFour additional morbiditiesFive additional morbidities

Age was categorized into four groups: (a) 18–39 years, (b) 40–59 years, (c) 60–79 years, and (d) 80 and over. Sex was categorized as male or female. We grouped occupation into different categories reported by the subjects themselves (not listed here).

## 4. Statistical Analysis

After a baseline characteristics exploratory analysis. We conducted to a chi-squared analysis to confirm deviations between surviving and no surviving patients. Further, we did a logistic regression test of hospitalized cases for COVID-19 to find out the influence of each covariate and the effect of prognostic factors on death. For the statistical analysis, we used the statistical package Stata version 15.1 (Stata Corp, 4905 Lakeway Drive, College Station, USA).

## 5. Results

Fifty-three percent of hospitalized patients with COVID-19 survived.Regarding the length of hospital stay, the median was seven days (0–99).The most common multimorbid profiles were diabetes/hypertension (20.14%), hypertension/obesity (9.25%) and diabetes/hypertension/obesity (5.0%), respectively.

Characteristics of the study population were stratified by age groups as shown in Table 1. Noticeably, the higher proportion of hospitalized cases corresponds to men (61.7%).

The logistic regression model from Table 2 shows major effects on survival outcome were attributed to intubation (odds ratio (OD) 8.601), age (OD 1.95), sex (OD 1.395), and chronic renal failure (OD 1.223). An important observation is the major effect of intubation.

Patients 18–39 were least affected, as showed by the lowest death rate; percentages can be observed in Table 3.

The most common single morbidities were hypertension (38.59%) diabetes (34.45%) and obesity (22%); multimorbid profiles were diabetes/hypertension (20.14%), hypertension/obesity (9.25%), diabetes/hypertension/obesity (5.0%), respectively. Less common were diabetes/hypertension/chronic kidney disease (CKD) (3.17%) and diabetes/cardiac disease (2.05%).

Hospitalised patients had different survival rates, according to the number of comorbidities. However, the presence of different numbers of patients in each group does not allow for a conclusion. Groups with three, four and five comorbidities showed the highest percentage of deaths. The chi-squared test showed a significant deviation, suggesting there were more deaths than expected (See Table 4). 

Though 30% of patients had no comorbidities, those with one associate disease made up 25% of the population. Increasing the number of morbidities decreased the number of patients involved, following a Poisson distribution.

Ninety-nine percent of hospitalized patients with COVID-19 and multimorbidity demanded medical attention in less than 14 days from signs’ and symptoms’ onset, and 8.75% of cases required mechanical ventilation. Likewise, the time elapsed to virus detection in survivors of COVID-19 was 20 days, on average, from signs of onset, whereas in no survivors it was undetected until death. The number of patients receiving vaccines before hospitalization was insignificant. Within the survivors, 53 had an incomplete dose and only 13 had complete vaccination.

The longest detection time observed in survivors was 37 days [7], making it imperative to ask the general population to demand medical attention early when signs and symptoms appear, which might reduce the risk of complication or death, particularly among those with multimorbidity who are at higher risk.

Women showed greater survival (14.30%), as well as patients under 65 years (19.2%). Managers or other professionals, health workers, and employees had higher survival rate (27.5%, 25.94%, and 23.39%, respectively). 

Concerning morbidity, patients with only hypertension who were hospitalized for COVID-19 had the highest survival (19.19%). Contrasting with the lowest survival in the diabetes/hypertension/CKD (6.3%) and diabetes/obesity (5.8%) groups. 

Patients hospitalized for COVID-19 without multimorbidity had a higher survival rate (19.32%).

Hospitalized patients demanding attention in less than 14 days survived 14.3%. Mechanical ventilation was a breakthrough. Patients not requiring mechanical ventilation survived at a rate of 24.5%, and only 4.3% of those who required mechanical ventilation survived.

Prognostic factors defining death associated with COVID-19 in hospitalized cases with multimorbidity showed females had a lower risk, together with employees, students, and health workers who were better off than the unemployed. In comparison, those with an age greater than or equal to 65 years had a greater risk of complications and death from this disease, as shown in Table 3.

From 1741 patients admitted to ICU; 837 survived (48.07%), and female survival predominated. The multimorbid hospitalized cases due to COVID-19 mostly associated with the lower survival were hypertension/CKD (RR: 8.97, 95% CI 2.24–35.94), diabetes/hypertension/CKD RR = 1.77 (95% CI 1.47–2.13), diabetes/hypertension RR = 1.33 (95% CI 1.19–1.46), diabetes/hypertension/obesity RR = 1.32 (95% CI 1.19–1.54), and hypertension/obesity RR = 1.21 (95% CI 1.04–1.41). Other factors such as pursuing care for longer than or equal to 14 days and the use of mechanical ventilation were associated with lower survival rates in those hospitalized with COVID-19 infection with multimorbidity.

## 6. Discussion

Most of the papers published to date about SARS-CoV-2 infection were written during 2020; before the vaccines were available. The conditions in hospital settings for treating COVID-19 infection were very different then compared to in 2021, a time when the vaccines are available to the general population. This breakthrough drastically changed the outcomes in hospitalized patients. Our study collected information on hospitalized patients between March 2020 and April 2021, evidencing a minimal number (66) of surviving patients who had incomplete or complete vaccination schemes.

This study was conceived at a time when no effective therapy was available to counteract the virus. Health professionals attempted a variety of pharmacological alternatives with all hospitalized patients, but with no success. 

We aimed for this study to find out what characteristics of the population were relevant regarding contracting the disease and then being admitted to the hospital, and finally to identify what factors were decisive in survival, as no solid ground was available in the literature.

In our study, hospitalized patients were a combination of different ages and number of multimorbidities, where survival was marked by conditions such as hospital admission. Our survival rate was lower than other studies (53%). Chen et al. [9] reported the deleterious effect of aging in infected patients, who were more prone to complications and death during their hospitalization; we observed in our study that people over 60 failed to recover in greater proportions.

Another issue of concern in our study was concomitant diseases present at the time of admission.

Chronic noncommunicable diseases influenced survival; the most affected patients had hypertension, diabetes, obesity, and chronic kidney failure, as has been reported by other authors [8].

The study by Nijman et al. in the Netherlands identified a similar pattern of patients to ours, with higher age (HRCS 1.10, 95% CI 1.08–1.12), immunocompromised state (HRCS 1.46, 95% CI 1.08–1.98), and who used anticoagulants or antiplatelet medication (HRCS 1.38, 95% CI 1.01–1.88) and had higher fatality rates. They found no increased mortality risk in male patients, or those with high body-mass index (BMI) or diabetes. In our study, obesity alone was a common factor, but it acted as protective, and was not associated with death. 

The study by Bellan et al. [8], conducted in Italy, showed that variables such as age, a diagnosis of cancer, obesity, and current smoking status independently predicted mortality. Our study did not show a relationship between smoking and death rate due to COVID-19 infection, and cancer was not identified as an important factor for the outcome among our patients.

The observational study conducted by Chudasama in the UK [12] is in alignment with our data, as the prevalence of multimorbidity was more than double in those with severe SARS-CoV-2 infection (25%) compared to those without (11%), and clusters of several multimorbidities were more common in those with severe SARS-CoV-2 infection. The most frequent common clusters with severe SARS-CoV-2 infection were stroke with hypertension (79% of those with stroke had hypertension); diabetes and hypertension (72%); and chronic kidney disease and hypertension (68%). Multimorbidity was independently associated with a greater risk of severe SARS-CoV-2 infection (adjusted odds ratio 1.91 [95% confidence interval 1.70, 2.15] compared with no multimorbidity).

In our study, the most common clusters were diabetes/hypertension, diabetes/obesity, and diabetes/hypertension/obesity, and no cases of stroke were identified in our study, maybe due to the age factor.

The study by Oliveira et al. [18] analysed 131 patients admitted to the ICU in Florida. They reported an overall hospital mortality and mechanical ventilation (MV)-related mortality of 19.8% and 23.8%, respectively. After excluding hospitalized patients, the ICU and MV-related mortality rates were 21.6% and 26.5%, respectively. In our study, 1696 patients were admitted to ICU. Over 50% had a fatal outcome (876), much higher than in Oliveira’s study, reflecting the surmounted hospital capacity in the region.

The study by Woolford et al. [19], demonstrated that 4510 participants tested for COVID-19 (positive = 1326, negative = 3184). Additionally, 497,996 participants were not tested. Compared to the non-tested group, after adjustment, COVID-19-positive participants were more likely to be frail (OR = 1.4 [95%CI = 1.1, 1.8]), which may have been related to age too. The population from Woolford is older than ours, at 57 years on average. 

Wolff et al. [20] identified conditions and comorbidities that were connected to a poor state of health. Among these were high age, obesity, diabetes, and hypertension, which are all risk factors related to severe and fatal disease courses.

Furthermore, severe and fatal courses are associated with organ damage, mainly affecting the heart, liver, and kidneys. Coagulation dysfunctions could play a critical role in organ damage. In our study, no cases of coagulation dysfunction were identified.

Dominguez-Ramirez et al. [21] confirmed the findings that chronic kidney disease (CKD) had the highest Relative Risk (RR) for COVID-19 fatality, followed by diabetes and immunosuppression, which in turn had higher RR than obesity or hypertension as single comorbidities. The combination of diabetes/hypertension with or without obesity had RR as high as CKD as a single comorbidity (>3 in <60-year-olds). Notably, the RR of comorbidities decreased with age, tending to reach a value near one after age 60, suggesting that comorbidities increase COVID-19 fatality in Mexico mostly in young and middle-aged adults. Our analysis suggests that advanced metabolic disease, marked by multimorbidity (more than one chronic disease per individual) or diabetes before age 60, contribute to the younger age of COVID-19 fatalities in Mexico. In Hidalgo, the observed survival rate of 24.5% at 40 days of hospitalization is within the range estimated by Ferroni et al., whose survival curves showed a death rate of 22% of patients in the first 14 days of hospital admission and 27.6% at 30 days [5].

Regarding sex, women were the best survivors; this finding is like Ferroni et al. [5]. Concerning age, patients under 60 years showed they were better fit to survive, as was identified in this study too, showing that the younger population had a greater survival to infection [5]. The relative risk increased in concordance with an increment of age; Salinas-Aguirre et al. and Parra-Bracamontes [22,23] also reported this observation.

Pre-existing disease conditions and the risk of worsening can also predict survival, for instance, via awareness. The perception of health can explain the lower possibility of awareness of abnormal symptoms and the days elapsed to demand care, which is consistent with identified factors in the study. Patients with higher-rank positions showed to be aware of signs earlier, which acted as a protective factor, followed by greater survival than the unemployed. Takagi collected data from 180 countries [24] to build up a meta-analysis, and found that the slope (coefficient) of the metaregression line for COVID-19 prevalence in the multivariable models was significantly negative for population ages 0–14 (−0.0636; *p* = 0.0021) and positive for obesity prevalence (0.0411; *p* = 0.0099), which would indicate that the COVID-19 prevalence decreases significantly as the proportion of children increases and vice versa as the proportion of the obese increases. 

The effect of multimorbidity is less associated with severe symptoms or deaths during the first months of expanding COVID-19 in the world. It has been reported that 15% of the cases with serious infection required hospitalisation and oxygen therapy. Only five percent of patients were in critical condition with complications such as respiratory failure, septic shock, pulmonary thromboembolism, or multiorgan failure, mainly due to kidney derangements [7]. Our study did not record cardiovascular complications of hospitalized patients; therefore, we are not in a position to speculate on this matter.

The study by Zhou [25] et al. included 191 patients, where 137 were discharged, and 54 died in hospital. Ninety-one (48%) patients had comorbidity, with hypertension being the most common (58 [30%] patients), followed by diabetes (36 [19%] patients) and coronary heart disease (15 [8%] patients).

In our study, a greater number of multimorbidity cases required hospital care (21.2%). In addition, a higher proportion required mechanical ventilation (12.6%), related to late seeking of medical care, which enabled quick progression of the disease, and admission to hospital units under worse clinical conditions. Aligned with Zhou, in our study hypertension was the most common morbidity present in COVID-19 hospitalized patients.

In the study by Pantea Stoian [26] et al., their findings indicate that male sex, hypertension, diabetes, obesity, and chronic kidney disease were most frequent factors in COVID-19 fatalities, that the burden of disease was low, and that the prognosis for 1-year survival probability was high in their 432-patient sample. In contrast, our study recorded that those fatalities were more related to hypertension but not to chronic kidney disease.

Factors identified as having a survival-reduction effect in hospitalized cases with COVID-19 were chronic diseases (mainly diabetes, hypertension, and obesity) [7,10,11], as shown by the influence of diabetes/hypertension/COVID-19 multimorbidity causing lower survival. This new viral infection is consolidated as a disease of high occurrence, and survival after infection is determined by the number of concomitant diseases present in a single individual [11].

One additional report by Gaipov et al. [27] identified a 24% higher risk of death in males than females and older patients compared to younger ones. In addition, patients residing in rural areas had a 66% higher risk of death than city residents and being treated in a provisional hospital was associated with 1.9-fold increased mortality compared to those treated in infectious disease hospitals.

In Denmark, Reilev et al. [28] identified 11,122 SARS-CoV-2 polymerase chain reaction-positive cases, of whom 80% were community-managed, and 20% were hospitalized. Thirty-day all-cause mortality was 5.2%. Age was strongly associated with fatal disease {odds ratio [OR] 15 [95% confidence interval (CI): 9–26] for 70–79 years, increasing to OR 90 (95% CI: 50–162) for 90 years, when compared with cases in those aged 50–59 years and adjusted for sex and number of comorbidities. Similarly, the number of comorbidities was associated with fatal disease [OR 5.2 (95% CI: 3.4–8.0), for cases with at least four comorbidities vs. no comorbidities] and 79% of fatal cases had at least two comorbidities.

In our study, the percentage of fatalities aligned with the number of comorbidities affected by age (See Table 4).

The most common observed chronic diseases (diabetes, hypertension, obesity, CKD, COPD) may be related to their high prevalence among the elderly. A disadvantage was posed between survivors and no survivors in the older group, who presented up to three comorbidities simultaneously. This fact decreased survival chances (diabetes/obesity and diabetes/hypertension/CKD). However, in the case of obesity, it is known that a higher prevalence among younger groups defines a higher percentage of survival compared to other comorbidities related to the patients’ age [6].

The report by Ioannou et al. [29] came to similar conclusions. In their study, in patients who tested positive for SARS-CoV-2, their characteristics that were significantly associated with mortality included older age (eg, ≥80 years vs. <50 years: adjusted hazard ratio (aHR), 60.80; 95% CI, 29.67–124.61), high regional COVID-19 disease burden (eg, ≥700 vs. <130 deaths per 1 million residents: aHR, 1.21; 95% CI, 1.02–1.45), higher Charlson comorbidity-index score (eg, ≥5 vs. 0: aHR, 1.93; 95% CI, 1.54–2.42), fever (aHR, 1.51; 95% CI, 1.32–1.72), dyspnoea (aHR, 1.78; 95% CI, 1.53–2.07), and abnormalities in certain blood tests, which exhibited dose–response associations with mortality, including aspartate aminotransferase (>89 U/L vs. ≤25 U/L: aHR, 1.86; 95% CI, 1.35–2.57), creatinine (>3.80 mg/dL vs. 0.98 mg/dL: aHR, 3.79; 95% CI, 2.62–5.48), and neutrophil-to-lymphocyte ratio (>12.70 vs. ≤2.71: aHR, 2.88; 95% CI, 2.12–3.91).

In our study, we were not able to estimate the Charlson comorbidity-index score. However, our main clinical features fit with those from Ioannou [29].

Likewise, whether it is a history of uncontrolled metabolism alone or the multisystemic conditions directly caused by the SARS-CoV-2 virus in the body, particularly mechanisms linked to the triggering of systemic pro-inflammatory cytokine response, the induction of procoagulant factors, and the hemodynamic changes and metabolic factors facilitating the virus entrance [11], different complications can be generated, and survival can be modified.

However, in current societies, we have disregarded that multimorbidity is the new normal, as Eck quotes [30]. It accompanies the intake of polypharmacy in the elderly and those not so elderly, with all their untoward consequences. Multimorbidity strongly correlates with prolonged medication use. About half of older adults in richer countries are taking five or more medications [30].

According to Ecks [30], the pharmaceutical industry promotes the regular consumption of five or more medications as necessary for maintaining health. For the industry, chronic polypharmacy is recommended as the new normal because multimorbidity is now normal. Even a decade ago, people in richer countries were on 9–13 prescription drugs in any given year.

Barnett et al. [31] found in their study that 42.2% (95% CI 42.1–42.3) of all patients had one or more morbidities, and 23.2% (23.08–23.21) were multimorbid. Although the prevalence of multimorbidity increased substantially with age and was present in most people aged 65 years and older, the absolute number of people with multimorbidity was higher in those younger than 65 years (210,500 vs. 194,996).

The current SARS-CoV-2 pandemic has revealed the burden of chronic diseases in modern society.

Although the number of patients was adequate, there were limitations to this study. For instance, no lab reports other than the PCR tests were available. Another limitation was the retrospective collection of information, which always depends on the person responsible for entering the data. The analysis was carried out with the existing information which was sometimes insufficient. 

## 7. Conclusions

This case study, similar to many others published in 2020, was under the limitation of a lack of available therapies, but there are many coincidental findings in the population. To conclude, multimorbidity reduces the survival of hospitalized COVID-19 patients by increasing the risks of clinical interrelationships among the coexistence of multiple diseases in the same individual. Thus, prevention strategies should be strengthened to avert infections coexisting with multimorbidity, as, if neglected, the probabilities of complication and death are high.

## Figures and Tables

**Table 1 healthcare-09-01423-t001:** Demographic Characteristics at Baseline.

Age Groups	18–39	40–59	60–79	80–over
Sex				
Male n (%)	1392 (59.7)	4143 (62.49)	3614 (59.41)	761 (56.96)
Female n (%)	939 (40.2)	2487 (37.51)	2469 (40.59)	575 (43.04)
Obesity				
No n (%)	1743 (74.77)	4832 (72.88)	4973 (81.75)	1178 (88.17)
Yes n (%)	587 (25.18)	1793 (27.04)	1107 (18.20)	156 (11.68)
Hypertension				
No n (%)	2004 (85.97)	4610 (69.53)	2901 (47.69)	538 (40.27)
Yes n (%)	326 (13.99)	2016 (30.41)	3176 (52.21)	796 (59.58)
Diabetes				
No n (%)	2084 (89.40)	4465 (67.35)	3358 (55.20)	845 (63.25)
Yes n (%)	245 (10.51)	2162 (32.61)	2718 (44.68)	488 (36.53)
Cardiac Disease				
No n (%)	2294 (98.41)	6457 (97.39)	5700 (93.70)	1190 (89.07)
Yes n (%)	35 (01.50)	169 (02.55)	379 (06.23)	144 (10.78)
Chronic Renal Dis				
No n (%)	2121 (91.1)	6273 (94.7)	5659 (93.2)	1250 (93.8)
Yes n (%)	208 (08.9)	354 (05.3)	419 (06.8)	84 (06.2)
Diab + Hypert				
No n (%)	1833 (96.06)	3541 (76.42)	2044 (52.31)	430 (53.02)
Yes n (%)	75 (03.93)	1647 (35.54)	1863 (47.68)	381 (46.97)
Diab + Obesity				
No n (%)	2256 (96.8)	5538 (83.6)	4220 (69.4)	955 (71.5)
Yes n (%)	75 (03.2)	1.092 (16.4)	1863 (30.6)	381 (28.5)
Hypert + Obesity				
No n (%)	2241 (96.2)	6008 (90.7)	5392 (88.7)	1224 (96.2)
Yes n (%)	90 (03.8)	622 (09.3)	69 (11.3)	112 (08.3)
Diab + Hypert + Obes				
No n (%)	2304 (98.9)	6307 (95.2)	5665 (93.2)	1287 (96.4)
Yes n (%)	27 (01.1)	323 (04.8)	418 (06.8)	49 (03.6)

See Appendix A. Diab = Diabetes, Hypert = Hypertension, Obes = Obesity, Chronic Renal Dis = Chronic Renal Disease.

**Table 2 healthcare-09-01423-t002:** Logistic regression test with death as dependent variable and main factors as covariates.

Death by COVID-19	Odds Ratio	Z Value	*p* > |Z|
sex	1.395	9.38	0.000
diabetes	1.062	3.18	0.001
age	1.958	28.68	0.000
obesity	1.062	2.87	0.004
chronic renal failure	1.223	5.68	0.000
cardiac disease	0.892	−2.72	0.007
COPD	1.010	0.20	0.778
hypertension	1.088	4.38	0.000
intubation	8.601	28.42	0.000
constant	0.005	−36.90	0.000

LR chi2(9) = 2361.56; Prob > chi2 = 0.0000.

**Table 3 healthcare-09-01423-t003:** Death rate by age group in a series of patients from Hidalgo, Mexico.

Age Group (Years)	Death Rate (%)
18–39	19.95%
40–59	36.65%
60–79	54.9%
80 and over	60.10%

See Appendix A.

**Table 4 healthcare-09-01423-t004:** Association between deaths and number of multimorbidities in a series of patients from Hidalgo, Mexico.

Number of Deaths by Number of Comorbidities							
	0	1	2	3	4	5	Total
Survived	3237 (64.8%)	3097 (57.5%)	2026 (51.6%)	949 (50.9%)	204 (45.6%)	47 (50.5%)	9575 (57.2%)
Died	1761 (35.2%)	2288 (42.5%)	1901 (48.4%)	916 (49.1%)	243 (54.4%)	46 (49.5%)	7163 (42.8%)

Pearson chi2(5) = 224.0029; Pr = 0.000. See Appendix A.

## Data Availability

Not Applicable.

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
