# Peer review of "Survival of COVID-19 with Multimorbidity Patients"

_healthcare, 2021, doi:10.3390/healthcare9111423_

Round 1

Reviewer 1 Report

Thanks for your nice work. I would prefer to classify the age of the studied population into 3 or 4 groups not only above or below 65 year. It is important to know which age group in more affected. This can help in designing the vaccination program for each group.

I hope in your next research work to study the multimorbidity with new variants of COVID-19 virus.  

Classification of individuals according to multimorbidity in the medical management of COVID-19 patients is important to determine the possible etiological models and to define patient triage for hospitalization. Also the identification of non-infected individuals with high-risk ages and multimorbidity patterns serves to define possible interventions of special management.

There are many published paper about the relation between multimorbidity and the severity of COVID-19 disease and all mentioned that multimorbidity reduced the survival length from COVID-19 infection in hospitalized patients.

This  paper can give a clue  about the situation in Mexico population who are of different ethnic comparing with other published studies.

Author Response

1.- We modified the age category according to your suggestion of 4 categories.

2.- During the time period the sample was collected our lab was about to initiate with variants determination. A further analysis will include the variants analysis

3.- We increased our references on multimorbidity in the text

Author Response

Dear Reviewer

Than you for taking your time to review or manuscript 

We did the a second global review and put special emphasis on improving the methods and results presentation 

Reviewer 3 Report

Some points should be considered in the manuscript.

  1. Introduction, the first paragraph should be referenced.
  2. The Title should be specific, and the audience should be global. I suggest adding more in the Introduction to make it more global. I think the study is specific to Mexico only.
  3. How did the authors come up with the groupings in Materials and Methods. It should be thoroughly explained.
  4. Results are more comprehensive if tabulated than stating it.
  5. Where is figure 1?
  6. Some Tables in the Supplementary Information are relevant. I think it would be better if transferred in the manuscript. Summarize and present it well.
  7. Make the Discussion more informative and make it global and inject Hidalgo data. I was reading a national issue than a global issue here.

Author Response

Responses to Reviewer 3 Comments

Point 1

Introduction, the first paragraph should be referenced.

It has already been referenced

Point 2

The Title should be specific, and the audience should be global.

The title has been modified to shorter

I suggest adding more in the Introduction to make it more global. I think the study is specific to Mexico only.

The Introduction has been modified to a global perspective

Point 3

How did the authors come up with the groupings in Materials and Methods? It should be thoroughly explained.

The explanation for grouping construction has been extended

The groups were modified to no comorbidities, 1, 2, 3, 4, 5 6 comorbidities

Point 4

Results are more comprehensive if tabulated than stating it.

Results have been changed to tables

Point 5

Where is figure 1?

Figure 1 has been included in the manuscript

Point 6

Some Tables in the Supplementary Information are relevant. I think it would be better if transferred in the manuscript. Summarize and present it well.

Tables from the Supplementary information are being added to the manuscript

Make the Discussion more informative and make it global and inject Hidalgo data. I was reading a national issue than a global issue here.

The discussion was changed

In addition, we are sending the manuscript to English edition

Round 2

Reviewer 3 Report

  1. Line 35: double dot after the sentence.
  2. Spellcheck and space check.
  3. I think Introduction could be better if just 2 paragraphs. Pool all paragraphs from lines 61 to 124. And, lines 125 should be your study and your limitations of the study, place it was conducted.
  4. Materials and methods, put place of a study.
  5. The content is good but the authors need to introduce Mexico in the Introduction so that they could plot their data well. Also, in Data collection, line 140, please put country or Mexico here.
  6. Results should be presented well. The tables should be improved. Tables 2 and 4 should be simplified. Table 3 is good but improve title by including place, i.e. Hidalgo Mexico. Table 5 should be deleted. I think the authors should put important data only. Please improve presentation of results.
  7. Figure 1. Is this your data? If not, delete this. If you want to use this, please get copyright permission.
  8. Please put your results only and discuss well, correlate with other literature.
  9. Discussion is okay but needs organization based on results.

Author Response

Please see attached manuscript file.

This manuscript is a resubmission of an earlier submission. The following is a list of the peer review reports and author responses from that submission.